# Method to Determine the Far-Field Beam Pattern of A Long Array From Subarray Beam Pattern Measurements

**DOI:** 10.3390/s20041236

**Published:** 2020-02-24

**Authors:** Donghwan Jung, Jeasoo Kim

**Affiliations:** Department of Ocean Engineering, Korea Maritime and Ocean University, Busan 49112, Korea; wjdehdghkss@gmail.com

**Keywords:** beam pattern, subarray, discrete line array, array sonar, near-field

## Abstract

Beam pattern measurement is essential to verifying the performance of an array sonar. However, common problems in beam pattern measurement of arrays include constraints on achieving the far-field condition and reaching plane waves mainly due to limited measurement space as in acoustic water tank. For this purpose, the conventional method of measuring beam patterns in limited spaces, which transform near-field measurement data into far-field results, is used. However, the conventional method is time-consuming because of the dense spatial sampling. Hence, we devised a method to measure the beam pattern of a discrete line array in limited space based on the subarray method. In this method, a discrete line array with a measurement space that does not satisfy the far-field condition is divided into several subarrays, and the beam pattern of the line array can then be determined from the subarray measurements by the spatial convolution that is equivalent to the multiplication of beam pattern. The proposed method was verified through simulation and experimental measurement on a line array with 256 elements of 16 subarrays.

## 1. Introduction

Beam-pattern measurement is essential to verify the performance of an array sonar. However, such measurements for long arrays generally need a huge testing field or a complex setup [1,2]. It is difficult to measure the beam pattern in an acoustic tank, especially for long arrays operating at high frequencies, because the acoustic tank usually does not meet the far-field conditions. To solve this problem, various studies has been conducted on methods of estimating beam patterns through near-field measurements. The earliest work in this direction was performed in the early sixties by Brown and Jull [3] for two-dimensional (2-D) cylindrical scanning, where the radiated field was expanded in terms of a series of radially expanding modes. Kerns [4] studied planar scanning, and Wacker proposed a method of extracting modal coefficients from spherical near-field measurements and a scheme using the fast Fourier transform (FFT) to calculate these coefficients [5,6]. Yaghjian examined the sources of major experimental errors that define the accuracy of a typical planar near-field measurement facility and summarized the principles and limitations of near, planar, circular, and spherical scanning methods [7]. Appel–Hansen provided a detailed description of planar, cylindrical, and spherical scanning in [8]. Ludwig used spherical-wave expansions to study the transformation between near-field and far-field data [9]. Recently, research using the spherical harmonic solution to the wave equation to transform the desired near beam pattern into a far-field beam pattern [10,11] and improve the method of transforming the conventional near-field scanning result into a far beam pattern has been studied [12,13,14,15,16].

In this study, we present a method to effectively measure very large (or long) array beam patterns in near-field environments. To verify the effectiveness of the proposed method, simulations were performed on several beam patterns in near field environments. In addition, the beam patterns were estimated from the actual measured values using the proposed method. Because the proposed method is simpler than conventional methods, it has the potential of allowing reduction in measurement time.

The remainder of this paper is organized as follows. In Section 2, we describe the proposed method for measuring the beam patterns. It consists of a description of the basic beam pattern (Section 2.1), which is the basis of the proposed method, and a description of the proposed method (Section 2.2). Section 3 gives the results of the simulation and experimental results. Section 4 discusses the simulation and experimental results, and we conclude the paper in Section 5.

## 2. Materials and Methods

Near-field scanning can be applied to a very large (or long) array, for which the far-field distance becomes prohibitive regarding the achievable experimental test range or the dimension of the acoustic tank. This allows effective performance measurements of a large array to be achieved. We intended to measure the beam pattern in a limited space for performance evaluation through the –3 dB beam width of the array. However, as near-field scanning is very time-consuming [2,14], we propose a method based on measurement of subarrays to measure the beam pattern of a long line array in a limited space. The method proposed in this study divides the long discrete line array into subarrays with the same size (number of elements) and estimates the long array beam pattern from the subarray beam pattern.

### 2.1. Beam Pattern

A beam pattern represents the intensity variation of a beam as a function of its direction and the distance from its source. Beamforming is a technique used to send or receive signals over or from a specific direction. It can be achieved through spatial filtering by using a sensor array or through signal processing. The beam pattern of a discrete array can be determined by summing the products of the phase differences and the delay functions of all of the array elements by considering the constructive and destructive interference of signals at a specific angle [17]. As shown in Figure 1, when the same omnidirectional elements are uniformly arranged over a plane and there is no mutual coupling among elements, the beam pattern is given by
(1)$$B\left(\theta ,\phi \right)=\overset{M}{\underset{m=1}{\sum }}\overset{N}{\underset{n=1}{\sum }}\overset{L}{\underset{l=1}{\sum }}g\left(x,y,z\right)\cdot exp\left[\left(m-1\right)\psi_{x}+\left(n-1\right)\psi_{y}+\left(l-1\right)\psi_{z}\right]$$
where angles θ and ϕ are as described in Figure 1; m, n, and l correspond to values along axes M, N, and L in Figure 1, respectively; g(x,y,z) is the aperture function; and ($\psi_{x}$,$\;\psi_{y}$,$\;\psi_{z}$) are expressed as
(2)$$\begin{array}{l} \psi_{x}=Kd_{x}\cos\theta \cos\phi \\ \psi_{y}=Kd_{y}\sin\theta \cos\phi \\ \psi_{z}=Kd_{z}\sin\theta \end{array}$$
with K being the wavenumber of the detected signal frequency and d the interval between elements [15,17,18]. Beam pattern B is defined for plane waves of sound because they satisfy the far-field conditions. 

### 2.2. Proposed Measurement Method

For a discrete line array whose measurement space does not satisfy the far-field condition, the proposed method for obtaining the beam pattern from near-field measurements divides the array into several subarrays, each satisfying the far-field condition. It is then possible to derive the beam pattern of the array by assuming each subarray is one element that contributes to the beam pattern. The aperture function of the line array consisting of subarrays is given by
(3)$$g\left(x\right)=\overset{N}{\underset{n=1}{\sum }}g_{n}\left(x-x_{n}\right)$$
where $x_{n}$ is the position of the subarray and $g_{n}$ is the aperture function of subarray n. If the aperture functions for all of the subarrays are the same and define a sampling function as in Equation (4), the aperture function of the array can be written in the convolution form shown in Equation (5), where ∗ denotes the convolution.
(4)$$D\left(x\right)=\overset{N}{\underset{n=1}{\sum }}\delta \left(x-x_{n}\right)$$
(5)$$g\left(x\right)=g_{n}\left(x\right)*D\left(x\right)$$

As convolution in spatial coordinates is multiplied by the angular area of the beam pattern, the total beam pattern can be obtained as
(6)$$B\left(\theta ,\phi \right)=B_{n}\left(\theta ,\phi \right)\times B_{D}\left(\theta ,\phi \right)$$
which corresponds to the beam pattern of the subarray multiplied by the beam pattern of the sampling function.

The order of proposed beam pattern estimation method of the array is shown in Figure 2.

## 3. Results and Discussion

### 3.1. Simulation

To verify the effectiveness of the proposed method, we first performed a simulation on a line array with 256 omnidirectional antenna elements at a design frequency of 455 kHz. As shown in Figure 3, the array was divided into 16 subarrays, each having 16 elements. The beam pattern for the far-field condition and that using the phase difference of each element at the near-field (r = 1.88 m, 2.44 m, 4.88 m) are shown in Figure 4. Although the ideal far-field beam pattern corresponds to a sine function, the beam pattern using the phase difference for the near-field in Figure 4 differs substantially from the far-field beam pattern. The beam pattern of each subarray and that of the sampling function required to obtain the total beam pattern using the proposed method for the far-field are shown in Figure 5. The beam pattern obtained using the proposed method on the results of Figure 5 are shown in Figure 6. It can be seen that the beam patterns estimated by the proposed method in Figure 6 are all the same regardless of the distance of the simulation. The calculated and simulated beam patterns agree, suitably, confirming the correctness of the proposed method. The comparison of the beam pattern obtained from the near-field phase difference using the proposed method and the far beam pattern is shown in Figure 7, which shows that it is effective to estimate the beam pattern by applying the proposed method to the near-field measurement.

### 3.2. Near-Field Measurement Experiments

To obtain a valid beam pattern and further verify the proposed method, we performed near-field measurement experiments similar to the simulated experiment on an array placed within a 4 by 4 by 5 m acoustic tank. For near-field measurements, the distance between the transmitter and receiver was 2.66 m and the measurement angle ranged from −50 to 50° at intervals of 1°. In addition, we considered a measurement interval from −10 to 10° for the main lobe with a resolution of 0.1° and a sampling frequency of 10 MHz. We obtained the beam pattern from near-field measurements to determine the performance through the −3 dB beam width of the array; thus, we limited the range of measurement angles. The signal used in the experiment has a central frequency of 455 kHz and a signal length of 10 wavelengths. The beam pattern of the subarray was obtained using the raw data and the signal passing through the band-pass filter. When the final beam pattern was obtained, the two beam patterns were compared. Figure 8 shows part of the signal obtained through the sub-array in the experiment. As these signals are raw data, noise is observed. The result of using the filter to remove the noise in the raw data is shown in Figure 9. The left side of each figure displays the result measured between −50° and 50°, and the right side shows the result measured between −10° and 10°.

Figure 10 illustrates the beam patterns obtained using data measured in the near-field for various subarrays. Although all the elements that comprise the array are nearly omnidirectional, this feature is not ideal, and there are measurement errors that cause a small difference in beam patterns among subarrays. To obtain the total beam pattern of the array through the proposed method, we assumed that the beam patterns of all subarrays are equal. Therefore, in this experiment, the average beam pattern from the measured subarrays (shown in Figure 11a) was used as the representative beam pattern, and the sampling beam pattern (shown in Figure 11b) was used to apply the proposed method.

The main lobe of the array beam pattern obtained from the proposed method that led to the results in Figure 11 are shown in Figure 12 (blue line). For comparison, the figure also shows the far-field beam pattern obtained from the simulation (red dashed line) for an array of 256 omnidirectional elements and simulation results (cyan line) for the same distance as the measured distance. The proposed method can suitably retrieve an array beam pattern from near-field measurements and can be employed to evaluate the performance of the array given the −3 dB beam width.

## 4. Discussion

We performed simulations and experiments to validate the proposed method. Simulations were performed for three near-field environments at 1.8 m, 2.44 m, and 4.88 m. The simulation results show that there is a difference between beam patterns in the near-field and far-field environments. In this study, the beam pattern estimated by the proposed method was exactly the same as the beam pattern in the far-field environment, thereby establishing the validity of the proposed method. We compare the beam pattern estimated from the measured experimental data using the proposed method with the simulation results under the same conditions as the measured experiment. The comparison results confirm the occurrence of a small error; however, this does not affect the validity of the proposed method. This small error may be interpreted as being caused by the measurement error and the directivity of the elements of the array.

## 5. Conclusions

We aimed to measure the beam pattern of an array to evaluate its performance over the −3 dB beam width. However, the limited available space for beam pattern measurements impedes the ability to satisfy the far-field condition of the array; when measuring beam patterns of arrays, the far-field condition is typically not satisfied, and the far-field measurements present various problems related to the transmission of sound waves.

We overcame this problem by using a method to obtain the beam pattern of the arrays through near-field measurements. The proposed method can reduce the measurement time compared with conventional methods such as near-field scanning; moreover, the process of beam pattern estimation is simple. A limitation of the proposed method is that the beam pattern of the sub-array in which the array is divided must be similar; however, because the beam patterns of the elements constituting the array are similar, the beam patterns between the sub-arrays are necessarily similar. Therefore, the beam pattern estimation of the array through the proposed method is valid. The validity, effectiveness, and accuracy of the proposed method were verified by simulation. Furthermore, we experimentally verified the proposed method by obtaining a successful beam pattern of an array through near-field measurements. This simple and effective method for estimating beam patterns is advantageous over conventional methods; furthermore, it avoids problems caused by limited measurement spaces, which fail to satisfy far-field conditions.

## Figures and Tables

**Figure 1 sensors-20-01236-f001:**
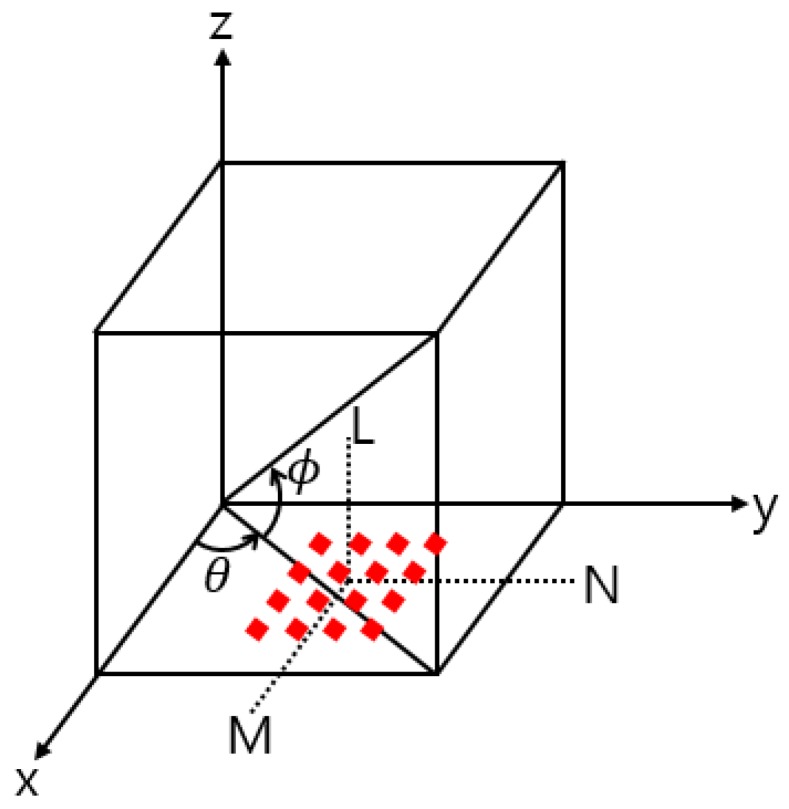
Geometry of a uniform rectangular array, where each element is a red diamond.

**Figure 2 sensors-20-01236-f002:**
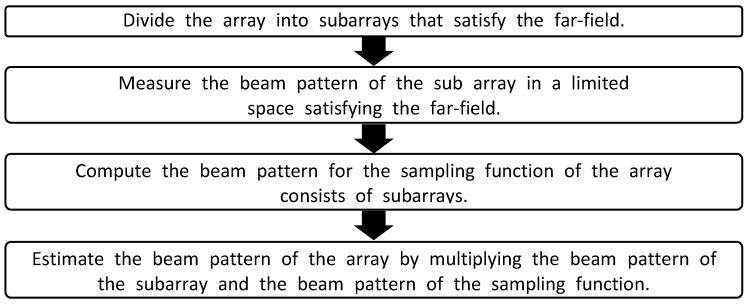
Flowchart of the proposed method.

**Figure 3 sensors-20-01236-f003:**
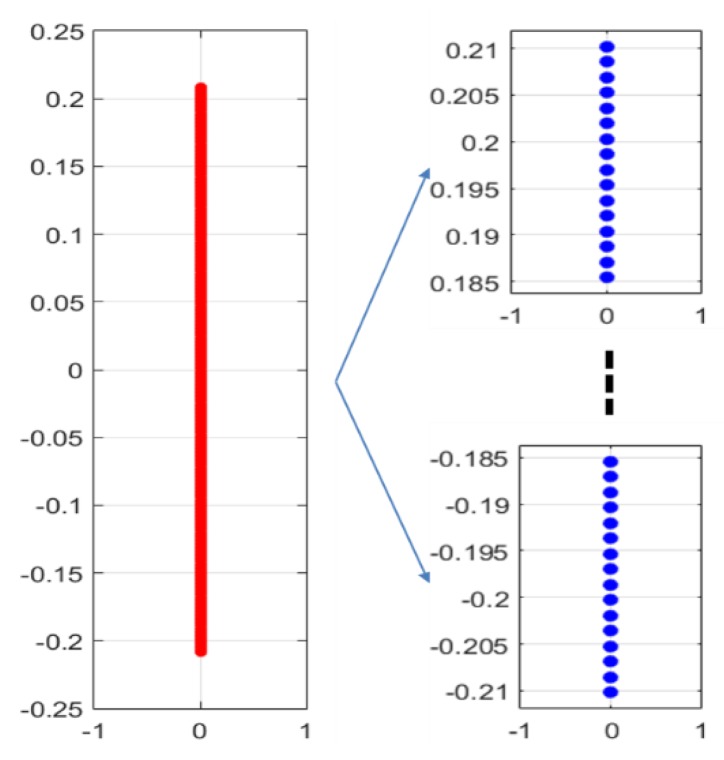
Array divided into 16 subarrays.

**Figure 4 sensors-20-01236-f004:**
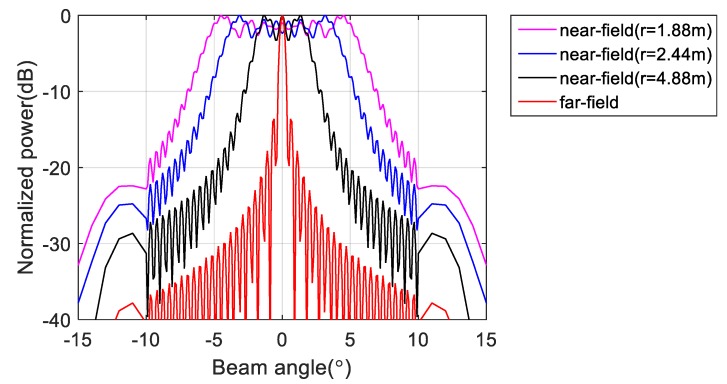
Beam pattern simulation for far-field and near-field (r = 1.22 m, 2.44 m, 4.88 m).

**Figure 5 sensors-20-01236-f005:**
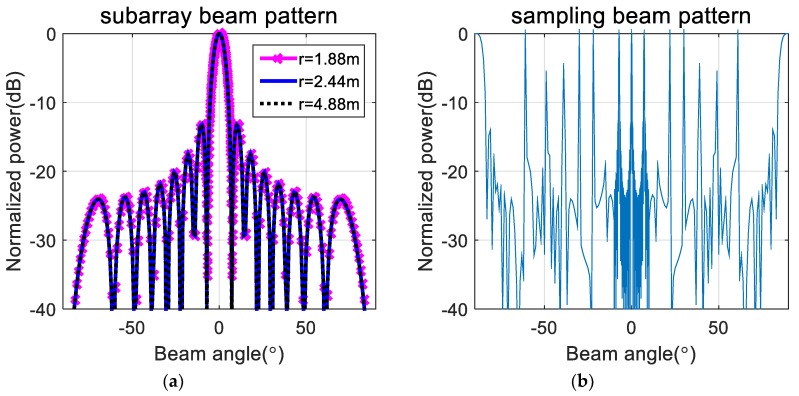
Beam pattern of subarray (**a**) and sampling function (**b**) in far-field.

**Figure 6 sensors-20-01236-f006:**
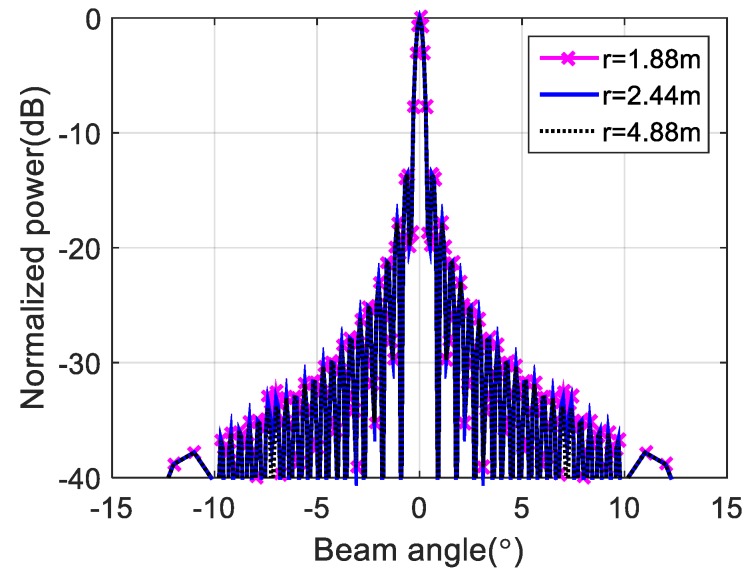
Beam pattern estimated by the proposed method.

**Figure 7 sensors-20-01236-f007:**
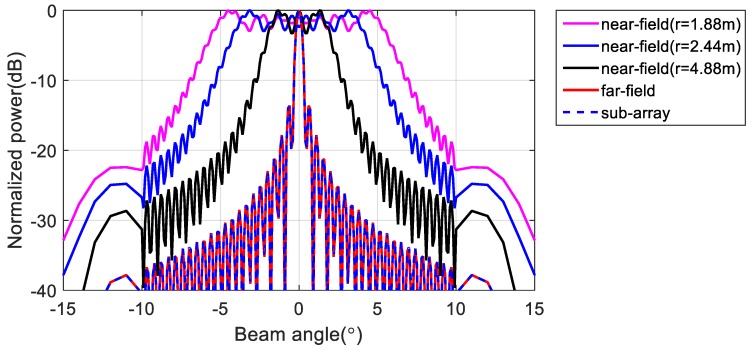
Beam pattern for the near field obtained by the proposed method and beam pattern for the far and near fields.

**Figure 8 sensors-20-01236-f008:**
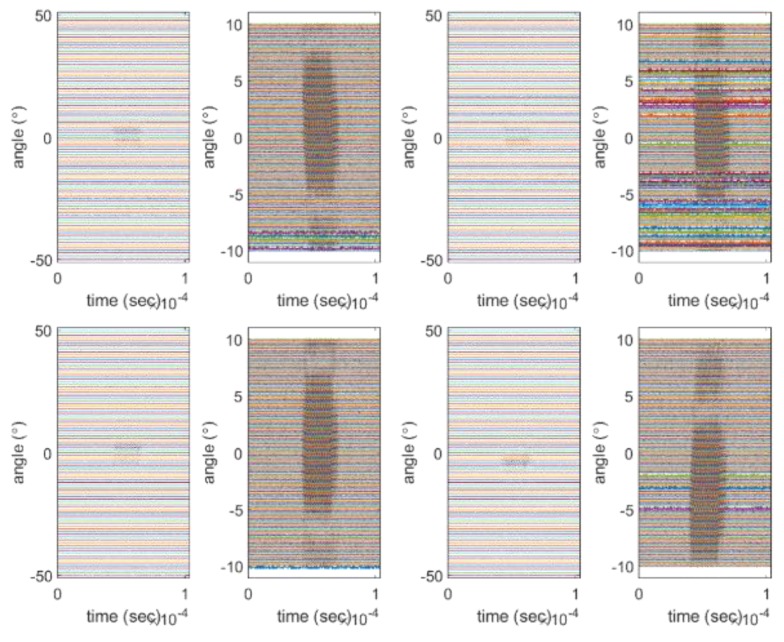
Example of raw data of the beam pattern of the sub-array: (**a**) ex 1; (**b**) zoomed ex 1; (**c**) ex 2; (**d**) zoomed ex 2; (**e**) ex 3; (**f**) zoomed ex 3; (**g**) ex 4; (**h**) zoomed ex 4.

**Figure 9 sensors-20-01236-f009:**
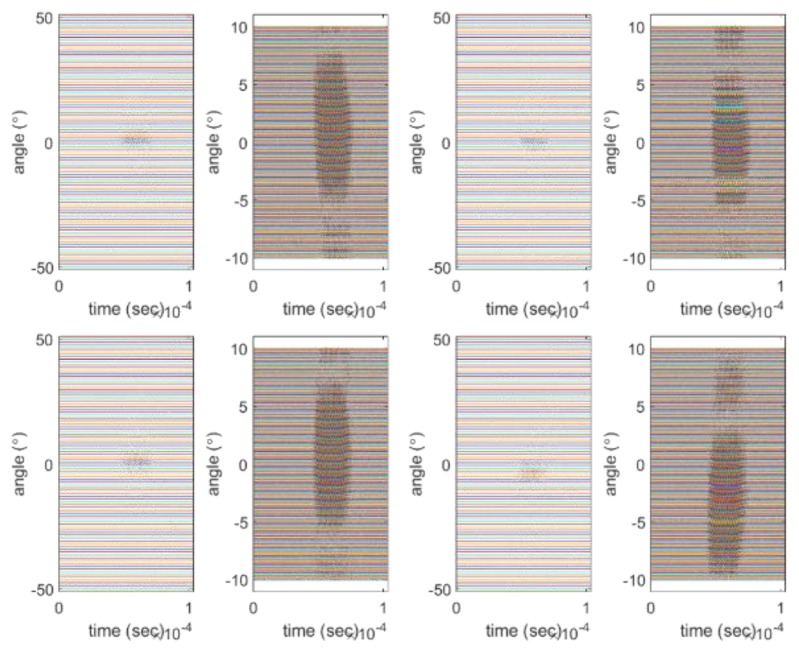
Example of filtered data: (**a**) ex 1; (**b**) zoomed ex 1; (**c**) ex 2; (**d**) zoomed ex 2; (**e**) ex 3; (**f**) zoomed ex 3; (**g**) ex 4; (**h**) zoomed ex 4.

**Figure 10 sensors-20-01236-f010:**
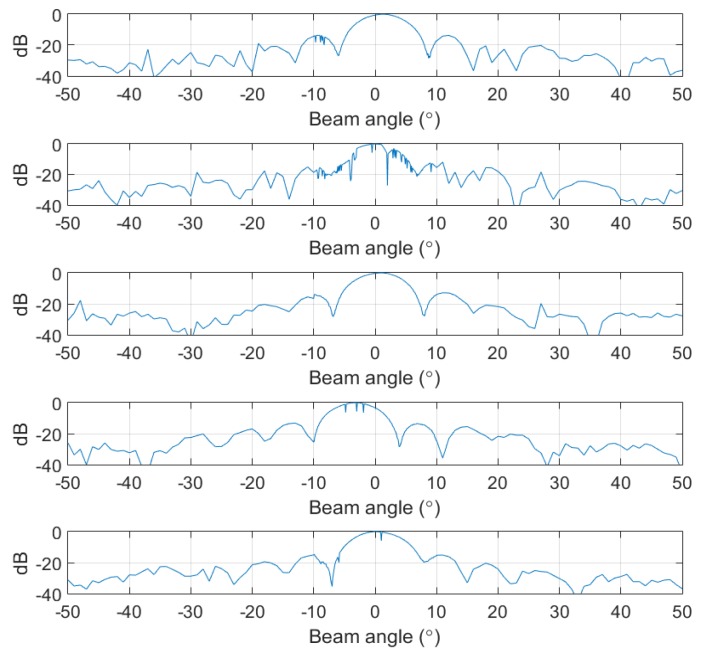
Examples of subarray beam patterns obtained from near-field measurements: (**a**) ex 1; (**b**) ex 2; (**c**) ex 3; (**d**) ex 4; (**e**) ex 5.

**Figure 11 sensors-20-01236-f011:**
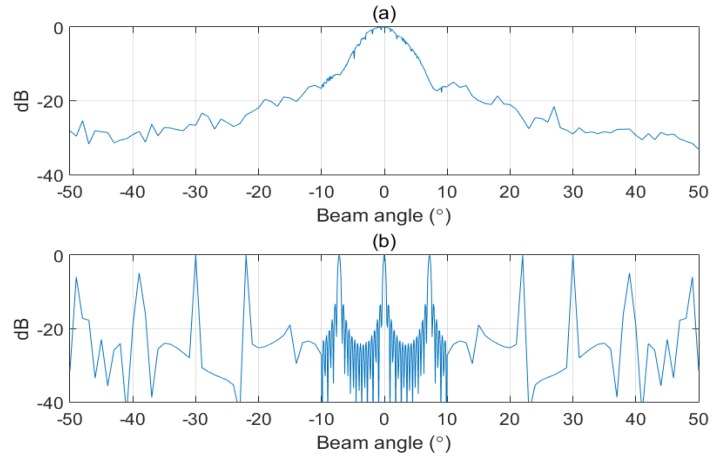
Subarray beam pattern to obtain total beam pattern using the proposed method. (**a**) subarray beam pattern obtained from near-field measurements and (**b**) sampling beam pattern.

**Figure 12 sensors-20-01236-f012:**
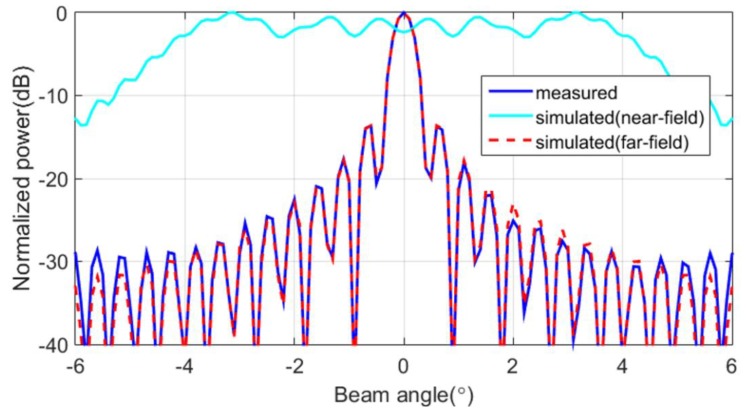
Beam patterns obtained from simulation and the proposed method.

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
