# Peer review of "Method to Determine the Far-Field Beam Pattern of A Long Array From Subarray Beam Pattern Measurements"

_sensors, 2020, doi:10.3390/s20041236_

Round 1

Reviewer 1 Report

The abstract should be revised. It should summarize the significant aspects of the entire paper and include: the overall purpose of the study and the research problem they investigated, as well as the significant findings and a summary of their interpretations and conclusions.

The “Introduction” should be a wide entrance to the subject. The introduction should be expanded as it only consisted of lines 25-32, then the authors started introducing their method (that part should be moved from the “Introduction” to the” proposed Model” section. The “Beam Pattern” section should be moved to the “Introduction”.

The paper structure and the flow of ideas should be revised.

Below are the major flaws in the paper, as it is not structured as a research paper and is missing the following:

Definition of the research problem that the authors are working on. There should be a separate section defining their research problem. A research paper is about resolving one well defined research problem. The paper’s contributions. The authors should enumerate the benefits that we could gain by adopting the authors’ proposed method. These contributions should be also explained in their conclusion. Related work and literature review. The proposed solution should be more elaborated. The authors should specify their input in the mathematical model. It might be a good idea for the authors to propose a diagram-model for their proposed method, and give it a name, and chose an acronym for it. The authors are not comparing the results they have obtained with their proposed method to the results of the existing ones. The authors should add a discussion and evaluation section on their findings and explain their significance. The authors did not specify their future work in their conclusion.

The article should be read word-to-word to correct sentence building and grammar. The use of the transition words, as well as the prepositions, should be revised. At many occurrences, there were missing links between the sentences.

The authors should review the equations and figures numbers.

Author Response

I agree with your comment.

In addition to adding the advantages of the proposed method to the Introduction and Conclusion sections, I have revised the manuscript for your comments.

I have attached a response file. Please check the response file for your comment.

Reviewer 2 Report

In this paper, the authors proposed a method to measure the beam pattern of a discrete line array in limited space based on the subarray method. Some suggestions are presented as follows.

The authors are suggested to highlight the main differences between the published works and the proposed work before introducing the details. In this simulation part, the authors are suggested to provide more results to verify the proposed methods.

Author Response

(The authors gave the same response as above.)

Reviewer 3 Report

As near-field scanning is very time-consuming, this article presents a novel method based on measurement of subarrays to measure the beam pattern of a long line array in a limited space. In the opinion of reviewer, it is very interesting and well organized. However, it has some problems that need to be solved further. Provided that the authors prepare a revision considering all issues outlined below, I would recommend the paper for publication in Sensors.

In introduction, the background of the article is not enough, more relevant content should be added, including some references and their comparison. The corresponding author's symbol is ignored in Line 4. Line 5 and 6 are not standard, the authors should check them carefully. There are only three keywords, which is not enough. This should be increased to at least five. Line 42 has nothing, the same problems can be found in Line 75, 95 and 112. There are two Section 2.1 in Line 44 and 59. All the formulas and symbols are fuzzy and the author needs to edit them again. There is a big gap on page 3. The authors should rearrange the article. The order of Figure 2, 3 and 4 should be corrected. Figure 7, 8, 9 and 10 have the same problems. The format of some references is not standard. The innovation of the paper is not clear enough, especially in the abstract and conclusion.

In order to qualify for publication in Sensors, the article must be improved according to the comments to the authors.

Author Response

I agree with your comment.

I have revised the manuscript for your comments, such as rearranging the manuscript, editing formulas and symbols, and writing the advantages of the proposed method.

Please check the answer file for your comment.

Round 2

Reviewer 1 Report

Most of my previous comments still stand as the authors have decided to pick and choose what to answer.

“The abstract should be revised. It should summarize the significant aspects of the entire paper and include: the overall purpose of the study and the research problem they investigated, as well as the significant findings and a summary of their interpretations and conclusions.

The “Introduction” should be a wide entrance to the subject. The introduction should be expanded as it only consisted of lines 25-32, then the authors started introducing their method (that part should be moved from the “Introduction” to the” proposed Model” section. The “Beam Pattern” section should be moved to the “Introduction”.

The paper structure and the flow of ideas should be revised.

Below are the major flaws in the paper, as it is not structured as a research paper and is missing the following:

Definition of the research problem that the authors are working on. There should be a separate section defining their research problem. A research paper is about resolving one well defined research problem. The paper’s contributions. The authors should enumerate the benefits that we could gain by adopting the authors’ proposed method. These contributions should be also explained in their conclusion. Related work and literature review. The authors have added a paragraph to the “Introduction” section. That is not the purpose of the “Introduction”. Its purpose is to have a wide entry to the subject, so that you could get the readers’ interest in your paper. The proposed solution should be more elaborated. The authors should specify their input in the mathematical model. The authors are not comparing the results they have obtained with their proposed method to the results of the existing ones. The authors should add a discussion and evaluation section on their findings and explain their significance. The authors did not specify their future work in their conclusion.

The article should be read word-to-word to correct sentence building and grammar. The use of the transition words, as well as the prepositions, should be revised. At many occurrences, there were missing links between the sentences.

The authors should review the equations and figures numbers.”

New Comments:

The authors should remove Figure 12 from the Conclusion and add it to a new “Discussions and Evaluation” section as previously requested, The authors should base their paper on more recent articles.

Author Response

Thank you for your review.

We reflected your comment as much as possible.

The response to your comment is in a file.
Please check the file.

Reviewer 2 Report

The authors have already adapted comments and I suggest to accept this paper after modifying format and checking English wording.

Author Response

Thank you for your review.

We checked your comment, the response to your comment is in a file.
Please check the file.

Reviewer 3 Report

Thank the authors for their efforts. The authors have adequately addressed my concerns in the review, and did a good job to revise and improve the paper. The paper now is suitable for publication in Sensors after minor spell checking and error correction.

Author Response

(The authors gave the same response as above.)
